# Immune-Escape Hepatitis B Virus Mutations Associated with Viral Reactivation upon Immunosuppression

**DOI:** 10.3390/v11090778

**Published:** 2019-08-24

**Authors:** Ivana Lazarevic, Ana Banko, Danijela Miljanovic, Maja Cupic

**Affiliations:** Institute of Microbiology and Immunology, Faculty of Medicine, University of Belgrade, Dr Subotica 1, 11000 Belgrade, Serbia

**Keywords:** hepatitis B virus, mutations, HBV reactivation, immune escape

## Abstract

Hepatitis B virus (HBV) reactivation occurs as a major complication of immunosuppressive therapy among persons who have recovered from acute hepatitis and those who have controlled chronic infection. Recent literature data emphasize the presence of a high degree of *S* gene variability in HBV isolates from patients who developed reactivation. In reactivated HBV, the most frequently detected mutations belong to the second loop of “*a*” determinant in HBsAg. These mutations were identified to be immune escape and responsible for vaccine- and diagnostic-escape phenomena. Their emergence clearly provides survival in the presence of a developed humoral immune response and is often associated with impaired serological diagnosis of HBV reactivation. The knowledge of their existence and roles can elucidate the process of reactivation and strongly highlights the importance of HBV DNA detection in monitoring all patients with a history of HBV infection who are undergoing immunosuppression. This review discusses the possible influence of the most frequently found immune-escape mutations on HBV reactivation.

## 1. Introduction

### 1.1. Natural History of HBV Infection and HBV Reactivation

It is estimated that more than two billion people around the world have been infected by hepatitis B virus (HBV) [1]. HBV infection remains a heavy health burden worldwide since chronic infection affects an estimated 248 million people and represents a significant risk for developing adverse outcomes, including cirrhosis, hepatic decompensation, and hepatocellular carcinoma [2]. It is estimated that around 686,000 people die each year from the complications of chronic hepatitis B.

HBV infection can be either acute or chronic and may range from asymptomatic infection or mild disease to severe or rarely fulminant hepatitis. The natural history of chronic hepatitis B (CHB) is exceptionally complex. It progresses nonlinearly through several recognizable phases [3,4]. The “immune-tolerant” phase is characterized by detectable HBeAg, high HBV DNA levels, often normal alanine aminotransferase (ALT) levels, and minimal or no liver necroinflammation or fibrosis. It is followed by an HBeAg-positive “immune-active” phase of active inflammatory disease where HBeAg is still positive, HBV DNA levels are still high, ALT levels are elevated, and moderate or severe liver necroinflammation and accelerated progression of fibrosis can be observed. The inactive “immune-control” phase (previously called the “inactive carrier phase”) follows successful seroconversion from an HBeAg-positive to anti-HBe state and is marked by undetectable or very low HBV DNA (<2000 IU/mL), normal ALT levels, and only minimal hepatic necroinflammatory activity and low fibrosis. HBeAg-negative (“immune escape-mutant”) active chronic hepatitis follows in some patients and is characterized by a lack of serum HBeAg with detectable anti-HBe antibodies, moderate to high levels of HBV DNA (>2000 IU/mL), with fluctuating ALT levels and necroinflammation, and more rapid progression to cirrhosis.

Although many basic problems regarding HBV immunopathogenesis have been solved, the phenomena of persistence, clearance, and recurrence of HBV have not yet been elucidated. As known, some patients spontaneously control the infection, while it becomes chronic in others. Virus-specific CD4 T-lymphocytes are thought to be essential for the control of the infection since they are required for both efficient B-cell/antibody and CD8 T-cell response. The epitopes mostly targeted by CD4 T-cells belong to the HBV core, but CD4 cells were also described to target envelope, polymerase, and X proteins to a minor extent [5,6,7]. In patients with acute or controlled chronic HBV infection, CD4 T-cell responses are more broadly directed and more vigorous, compared to patients with established active chronic infection [8,9]. Also, CD8 T-lymphocytes play an important role in the establishment of chronic infection by their cytolytic or non-cytolytic effector functions. CD8 T-cells targeting all HBV proteins have been identified [10]. However, during later stages of chronic infection, CD8 T-cells become functionally exhausted, which is thought to be driven by persistent antigen exposure but can also be the result of virus escape by generation of escape sequence variants. Thus, a lack of protective T-cell memory maturation and an exhaustion of HBV-specific T-cells are associated with chronic virus persistence. Upon prolonged suppression of HBV replication and a decline of antigen in patients treated with nucleos(t)ide analogues (NA), partial restoration of the T-cell function occurs, which indicates the significance of T-cell inhibition in chronic infection [11]. It is less known how the B-cell response and specific antibodies can contribute to viral control in an established chronic infection. Antibodies against envelope proteins (anti-HBs) possibly contribute to the control of viremia even in chronic infection, although they are not detected by standard laboratory assays since they form immune complexes [9].

Among persons who have recovered from acute hepatitis and those who have controlled chronic infection, in cases of severe immune system suppression, a significant viral replication can be established again and may lead to hepatitis flare and liver failure. At a conference in 2013 on “Reactivation of Hepatitis B” organized by the American Association for the Study of Liver Diseases, a definition of hepatitis B reactivation (rHBV) was proposed: Hepatitis B reactivation is defined as the abrupt reappearance of HBV (serum HBV DNA > 100 IU/mL) in the serum of a person with previously resolved infection or a marked increase of HBV replication (>2 log increase of serum HBV DNA from baseline level) in an immunosuppressed patient with previously stable chronic infection [12].

HBV reactivation usually occurs as a major complication of immunosuppressive therapy for a concomitant medical condition. It may be triggered by cancer chemotherapy (for hematological and other malignancies); immunosuppressive therapy (in bone marrow or solid organ transplant recipients); therapy for inflammatory, rheumatological, and other autoimmune diseases; and by direct-acting antiviral therapy for HCV infection [13]. It was also reported that any modulation to the immune system that may disrupt the interaction between the virus and host could be responsible for HBV reactivation. For instance, it was observed in pregnant women who experienced post-partum flares of hepatitis B infection [14]. Although rare, spontaneous reactivation of HBV infection has also been reported in elderly people with resolved HBV infection in the absence of known triggers for reactivation, probably as a result of age-related immunosuppression [15]. The risk of reactivation is associated with the specific immunosuppressive or immunomodulating drug and the duration of immunosuppression but also with the serological status of the patient and genetic variability of the virus.

So far, several virological and serological risk factors were associated with HBV reactivation. Candidates for chemotherapy and immunosuppressive therapy who are HBV DNA positive are considered to be at the highest risk for the development of viral reactivation and consequent hepatitis flare [13,16]. It was shown that HBsAg positive patients are eight times more likely to experience reactivation when compared to HBsAg negative/anti-HBc positive patients [17,18]. Also, in HBsAg positive patients, HBeAg positivity confers further risk of reactivation [17].

In recent years, the risk of HBV reactivation was observed in HBV/HCV co-infected patients undergoing anti-HCV direct-acting antiviral therapy (DAA) [19]. The explanation of this phenomenon is that HCV has a suppressive effect on HBV replication and that once this suppression is removed due to efficient DAA treatment, the environment is favorable for HBV reactivation [20,21]. On the other hand, the immunological changes occurring after HCV clearance can be related to this phenomenon rather than direct interference between HCV and HBV [22].

### 1.2. Molecular Virology and Genetic Variability of HBV

The human HBV is a prototypical member of the family *Hepadnaviridae*, which includes a variety of similar avian and mammalian viruses. The HBV genome is in the form of partially double-stranded circular DNA, which is contained in an icosahedric capsid, itself enveloped by a lipid bilayer bearing three different surface proteins. In virions, the genome is in a relaxed circular conformation (rcDNA) and attached to endogenous polymerase. The minus DNA strand is complete, but the plus strand has a gap of about 600 nucleotides [23].

The HBV life cycle starts with virion attachment, which has only recently been elucidated [24,25,26]. HBV initially binds, in the perisinusoidal space (or space of Disse) in the liver, to heparan sulfate proteoglycans [24] expressed on the surface of hepatocytes and then with high specificity to a bile receptor—the sodium taurocholate co-transporting polypeptide (NTCP)—and subsequently becomes internalized in endosomal vesicles [25,26]. After entry into the hepatocyte, the nucleocapsid is actively transported to the nucleus via microtubules. The small diameter of the capsid allows it to pass through the nuclear pores [27]. Upon entry into the nucleus, rcDNA is first converted to a fully double-stranded genome, which is then supercoiled to form covalently closed circular DNA (cccDNA). cccDNA binds to histones and other chromatinizing proteins to form a mini chromosome, which has a very long half-life and serves as the major transcriptional template for the virus [28]. This mini chromosome allows HBV to persist in the hepatocyte and determines significant clinical characteristics, including chronicity and reactivation, carcinogenesis, and the relative inefficacy of antiviral treatment.

From the minichromosome, genomic and subgenomic viral RNAs are transcribed, and are then translated into viral proteins in the cytoplasm. Three proteins (HBV core antigen, HBcAg; soluble hepatitis B e antigen, HBeAg; and polymerase, Pol protein) are translated from the longest mRNA transcript of 3.5 kb. Core protein and HBeAg are both derived from the precore/core precursor molecule. HBeAg is generated by maturation of the precursor molecule in the endoplasmic reticulum, where 19 *N*-terminal amino acids are cleaved off, and then by removal of the basic *C*-terminal tail during transport to the cell surface. This protein is secreted as soluble antigen and its role is probably to serve as a decoy to protect infected cells from an immune response [29]. Three shorter transcripts, the subgenomic mRNAs, are translated into surface envelope (HBsAg) and HBx proteins. The longest transcripts of 3.5 kb also serve as template for reverse transcription of the viral genome when they are called pregenomic RNAs (pgRNAs). In the cytoplasm, nucleocapsids are generated by the assembly of core proteins around pgRNA and polymerase. Binding of the polymerase to the packaging signal, epsilon, at the 5’ end of the pgRNA begins the process of nucleocapsid assembly. Reverse transcription occurs in newly generated nucleocapsids [30]. After completion of the minus DNA strand, pgRNA is degraded by the RNase H activity of the polymerase, after which a positive DNA strand is synthesized, and finally the genome is circularized to form rcDNA. The surface proteins are synthesized and processed in the endoplasmic reticulum. Capsids containing rcDNA are directed to multivesicular bodies to be enveloped and released from the cell by exocytosis [23,31].

During the genome replication, some of the newly synthesized rcDNA get recycled to the nucleus to maintain the cccDNA pool. In a natural HBV infection, there are more than 50 copies of cccDNA in a hepatocyte due to intracellular amplification [32]. There is evidence an infected hepatocyte will harbor cccDNA and replication intermediates, such as pregenomic RNA (pgRNA), indefinitely until its elimination [12,33]. Also, as a byproduct of viral replication, a double-stranded linear (dsl) DNA can be formed. This form of viral DNA can be a substrate for an unexpected event in the HBV viral life cycle—integration into host DNA by the process of recombination. This event occurs in approximately 1 in 10^3^ to 10^4^ infected cells in animal models [34].

Soon after the discovery of “Australian antigen” or HBsAg, it became apparent that the newly discovered virus was highly variable. Initially, based on the antigenic heterogeneity of HBsAg, four serotypes were identified: *adw*, *adr*, *ayw*, and *ayr*. Later description of additional subdeterminants of “*a*” meant the number of serological subtypes increased to 10: *ayw1*, *ayw2*, *ayw3*, *ayw4*, *ayr*, *adw2*, *adw3*, *adwq*, *adr*, and *adrq*− [35]. So far, based on a genome sequence divergence of more than 8%, 10 HBV genotypes (A–J) have been identified [36]. Newly discovered genotypes I and J are still not recognized by all, since they are considered to be recombinants of genotype C. Additional classification to subgenotypes was introduced for isolates of some genotypes (A, B, C, D, and E), based on a genome sequence divergence of 4% to 8%, and so far around 40 subgenotypes have been identified [37]. The genotypes, subgenotypes, and serotypes show distinct geographical distributions. The diversity reflected in genotypes/serotypes is the result of evolutionary drift of the viral genome as a consequence of a long-term adaptation of HBV to genetic determinants of specific host populations.

However, it soon became evident that HBV was also prone to variability that arose spontaneously during replication. These spontaneous variations are the result of a unique viral life cycle, which includes the activity of an error-prone reverse transcriptase, and also of a very high replicational rate. In chronically infected patients, the daily rate of de novo HBV production may reach 10^11^ virions [38]. The estimated mutation frequency is 1.4 to 3.2 × 10^−5^ substitutions/site/year, which is approximately 10-fold higher than for other DNA viruses [39]. Therefore, HBV exists as a a quasispecies population, in which eventually a predominant strain is selected by endogenous (host immune system) and exogenous factors (antiviral therapy and vaccination).

### 1.3. Biology of HBV Surface Antigen and Immune-Escape Mutations

The HBV surface antigen (HBsAg) is the major antigen of the viral envelope and comprises the regions involved in the viral attachment to the hepatocytes and the main epitopes recognized by neutralizing antibodies and T-lymphocytes. It is composed of three viral envelope proteins—large (LHB), middle (MHB), and small (SHB), all encoded by a single open reading frame—*S*-ORF. The *S*-ORF is divided by three start codons into the following domains: *PreS1*, *PreS2*, and *S* [40]. The three surface proteins are translated from 2 subgenomic mRNAs—LHB from 2.4 kb subgenomic mRNA, transcribed from *PreS1*, *PreS2*, and *S* domains; and MHB and SHB from 2.1 kb subgenomic mRNA, transcribed from *PreS2* and *S* domains. The transcription of 2.4 kb subgenomic mRNA is regulated by the *preS1* promoter and transcription of 2.1 kb subgenomic mRNA by the *preS2/S* promoter, which also belongs to the *PreS1* domain [41,42].

Since LHB is encoded by all three domains of the *S*-ORF, it has extra amino acids (108 or 119 aa depending on the genotype) at the *N*-terminus relative to MHB. Since LHB is abundant in virions and present in very low amounts in subviral particles, it is recognized as the primary ligand for the viral receptor. It has recently been suggested that a high specificity interaction occurs between the *N*-terminal 75 amino acids of the *PreS1* domain of the LHB and NTCP, a bile receptor on hepatocytes [25,26]. Besides mediating viral entry, LHB is also required for the binding of capsids and the assembly of virions before release from the cell [43,44]. During genome replication, capsids gain the ability to interact with envelope proteins and the *N*-terminal domain of LHB plays an important role in this interaction. Although the role of MHB in the HBV life cycle was considered an enigma, MHBs were found to influence the production of extracellular virions, which was a role previously assigned only to SHB [42,45]. Both LHB and MHB can activate transcription from selected promoters in transfected cells [46].

All three envelope proteins are partially *N*-glycosylated at asparagine 146 in the *S* domain [47]. MHB is *N*-glycosylated at asparagine 4 of the *pre-S2* domain [48]. This amino acid is not modified in the LHB protein because it remains on the cytosolic side of the ER membrane during protein synthesis. However, the myristoylation of glycine 2 in the LHB protein seems to play an important role in the infection process [49,50].

A balanced expression of envelope proteins appears to be vital for the HBV life cycle. The expression of the envelope proteins regulates the amplification of cccDNA in the nucleus [51,52,53]. It was found that the level of cccDNA increased when expression of the envelope proteins was ablated.

The *S*-domain of all three surface proteins contains various B- and T-cell epitopes important for inducing the host immune response. The central core of SHB, comprising amino acids 99 to 169, is exposed on the surface of virions and is involved in binding to anti-HBs antibodies. It is referred to as the major hydrophilic region (MHR) and contains a cluster of B-cell epitopes, called the “*a*” determinant, comprising amino acids 124 to 147 [35]. This most important antigenic determinant in envelope proteins is composed of two loops bounded by disulfide bridges between cys124 and cys137, and cys139 and cys147. The “*a*” determinant is highly conserved and found in all genotypes and serotypes of HBV.

When an amino acid sequence with an “*a*” determinant is altered, as a result of point mutation, deletion, or insertion within the *S* domain of the *S*-ORF, important changes with regard to immunity and protection from HBV infection may appear. Amino acid changes within the “*a*” determinant, arise from selection or natural variation and can lead to conformational changes, which can affect the binding of neutralizing antibodies with several possible consequences: (1) False-negative results by some commercial assays for HBsAg (occult hepatitis B); (2) evasion of anti-HBV immunoglobulin therapy; and (3) evasion of vaccine-induced immunity. All of them influence the antigenicity of HBsAg and are sometimes referred to as “immune-escape” mutations. It is important to note that these escape mutations can arise from host factors alone, without selection caused by vaccination or anti-HBV immunoglobulin therapy [54,55].

In most cases, HBsAg immune-escape mutants are due to missense mutations, often involving only one amino acid residue, rather than to insertions or deletions of multiple residues. The phenomenon of vaccine escape was first introduced in 1988 in a follow-up study of childhood vaccination, which revealed that vaccinated children with a strong antibody response to HBsAg could still become HBsAg positive [56]. The first described vaccine escape mutation associated with this phenomenon was substitution of a Gly residue at position 145 by an Arg residue (G145R) [57]. This mutation has since become the most widely reported vaccine-escape mutant, but reports of many other substitutions in the “*a*” determinant have since been associated with escape from vaccine-induced immunity: T116N, P120S/E, I/T126A/N/I/S, Q129H/R, M133L, K141E, P142S, D144A/E, and G145R/A [35,58,59].

Occult hepatitis B infection (OBI) is defined as the presence of HBV DNA in the liver (with detectable (usually <200 IU/mL) or undetectable HBV DNA in the serum) of individuals testing HBsAg negative by currently available assays [16]. In the majority of cases, it is caused by host-related factors (immunological and epigenetic), but sometimes OBI is associated by viruses harboring *S*-gene immune-escape mutations. This condition, usually described as “false” OBI, is characterized by serum HBV DNA levels comparable to those detectable in the overt infection and is believed to be the major cause of HBV transmission by blood transfusion [60,61]. A significant number of *S* gene mutations within and outside of the “*a*” determinant have been identified and associated with OBI as follows: Y100S, Q101R, P105R, T115N, T116N, G119R, P120L, R122P, T123N, C124R/Y, T126I/S, P127H/L, Q129P/R, M133T, Y134C, S136P, C139R, T140I, K141E, S143L, D144A, G145R/A, S167L, R169H, S174N, L175S, V177A, and Q181STOP [62,63,64]. In addition to missense mutations, it has been strongly suggested that deletions in the *Pre-S* region could play a significant role in OBI development since they can affect the expression, synthesis, and secretion of the *S* protein [65].

## 2. Mutations Associated with HBV Reactivation

### 2.1. The Implications of HBsAg Variability

The impaired balance between viral replication and immune control can be responsible for an increase of HBV replication in chronically infected patients or reactivation of inactive HBV in recovered patients in the setting of immunosuppression. The risk of HBV reactivation in patients receiving immunosuppression is associated with the specific immunosuppressive drug or class of drug prescribed, the duration of immunosuppression, and also with the patient’s virological and serological status.

Patients positive for HBsAg are eight times more likely to experience HBV reactivation than those with evidence of resolved infection [17,18]. However, reactivation is often reported in HBsAg negative/anti-HBc positive patients whose risk remains owing to the persistence of HBV in the form of cccDNA in hepatocytes and other tissues [12,33]. Also, the presence of anti-HBs antibodies was identified as a protective factor since it was shown that an undetectable anti-HBs level at the start of immunosuppressive therapy represented an increased risk for HBV reactivation [66]. Despite this, many reported HBV reactivation cases in patients positive for anti-HBs antibodies contribute to the hypothesis that immune-escape HBsAg mutations confer risk of HBV reactivation [67,68,69,70,71]. Accordingly, a few studies and numerous case reports have emerged in recent years that emphasize the high degree of *S* gene variability in reactivated HBV DNA [67,68,72].

In contrast to the fact that HBsAg positivity bears a much higher risk for HBV reactivation, in the majority of cases reporting an association of HBsAg mutations and reactivation, the patients were HBsAg negative prior to reactivation [68,70,73,74,75,76]. This may be due to a number of possible factors. The first reason may be a reporting bias since cases of reactivation in the setting of resolved infection always attract more interest, prompting scientists to look for possible causes and reassess prevention protocols. The second reason can be that the number of reactivations was caused by infection that was not truly resolved but occult chronic infection. There is strong evidence that the host’s immune surveillance plays an important role in the OBI development, which is why immunosuppression can trigger OBI reactivation with the subsequent reappearance of the serological profile of overt infection. Also, host epigenetic modifications, such as methylation of viral DNA and acetylation of histones, are often related with OBI [77]. Methylation of HBV DNA can alter HBV proteins, replication, and virion production, which may lead to OBI [78]. HBV replication is regulated by the acetylation of H3/H4 histones bound to viral cccDNA [79]. Finally, the absence of HBsAg in patients who later develop reactivation can be the consequence of HBsAg mutations. Many *S* gene mutations, previously associated with OBI, are reported in HBV isolates from patients with reactivated infection (Figure 1).

In all cases of HBV reactivation, it is likely that initially, viral replication had persisted at very low levels in the liver since resolution of acute infection or stabilization of chronic infection. This persistence of viral replication may be the consequence of immune-escape mutations since some of them provide survival in the presence of a developed humoral immune response. However, the weakening of the immune system during immunosuppressive therapy results in a gradual increase of viral replication, which in turn enhances the emergence of immune-escape mutants [73]. The viral quasispecies structure is shifted towards highly mutated variants [67]. The possibility of some HBsAg mutants contributing to an increase of replication has also been suggested.

#### 2.1.1. Mutations within “*a*” Determinant

The majority of HBsAg mutations, reported in reactivated HBV, resided in MHR, mostly within the “*a*” determinant (Figure 1). Not surprisingly, the most often found mutation within the “*a*” determinant was at position 145, which is the most widely studied vaccine-escape mutation. It was present as a glycine to arginine change (G145R) but also as a change from glycine to alanine (G145A). The G145R variant was found to gain infectivity in vitro despite changes in the antigenicity of the “*a*” determinant, which is why it preferentially emerges. The infectivity is explained by more efficient attachment to the heparan sulfate of this variant [91]. It was also shown by computational analysis that this mutation caused a considerable reduction in the immunogenic activity of the HBsAg through a conformational change in the HBsAg antigenic loops [92]. Non-vaccinated people can also harbor this mutation but only on a minor quasispecies population. It can only emerge to become the major viral population in patients in the face of immune pressure, usually vaccine-induced or prophylactic treatment of liver transplant patients with human anti-HBs immunoglobulins (HBIg). Accordingly, in the majority of reported cases of reactivation, this mutation was found in patients positive for anti-HBs antibodies at baseline and sometimes even after the onset of reactivation [66,68,69,73,81,83,84,85].

Other known immune-escape mutations within the “*a*” determinant were found isolated or together with a mutation at position 145, most notably those at positions 144, 134, and 142 (Figure 1). All mutations within the “*a*” determinant that are able to change the hydrophilicity, electrical charge, or acidity of the amino acid loops (those reported in patients with reactivation: 126, 127, 129, 131, 133, 134, 142, 143, 144, and 145) are responsible for changes in the loops’ structure and the antigenicity of HBsAg [93]. Mutations affecting positions of the disulfide bridges (position 124 reported in patients with reactivation) can change the confirmation of the loops. The pattern of immune-escape mutations may depend on the HBV genotype since some of the mutations are more likely to be found in certain genotypes. The example would be that G145R is mainly found in genotypes B, C, and D [94]. Also, the frequency and pattern of these mutations are probably dependent on the initial serostatus of the patient, with them more often found in HBsAg-negative/anti-HBs-positive patients. It was suggested as well that the pattern of the mutations could depend on the type of immunosuppressive regimen. Although HBsAg mutations were found in cases of reactivation caused by different immunosuppressive factors (transplantation, HIV, corticosteroid use, anti-HCV DAA therapy, age-related immunosuppression), those within the “*a*” determinant are thought to be essential for reactivation in patients undergoing treatment with anti-CD20 monoclonal antibodies (rituximab, ocrelizumab) [82]. Since these monoclonal antibodies target and destroy B-lymphocytes, the immune-escape mutations further favor reactivation probably by allowing the virus to escape an already impaired humoral immune response.

#### 2.1.2. Additional *N*-linked glycosylation Sites

The *S* gene point mutations, found among patients experiencing reactivation, could be associated with HBsAg negativity by introducing additional *N*-linked glycosylation sites (NLG) to HBsAg. *N*-linked glycosylation is required for crucial biological functions of many enveloped viruses and can provide advantages in viral survival and virulence [95]. The *S* domain of all three envelope proteins is partially *N*-glycosylated at position N146, and MHB at N4. The position 146 is within the “*a*” determinant, and it was shown that the removal of this site decreased the production of virion but without affecting the synthesis and stability of HBsAg [96]. Several aa sites within envelope proteins were reported to represent additional NLG sites and cause a decrease of HBsAg antigenicity and consequently OBI [64].

The additional NLG sites within the MHR of HBsAg were reported in patients with HBV reactivation by several different studies (Figure 2). In the study by Salpini et al. [67], 24.1% of patients with reactivation displayed additional NLG sites within the *S* gene of HBV, either as single-point mutations or patterns of two: T115N, ins115N, T123N, S113N + T131N, and ins114N + T117N. All reported NLG sites except the pattern S113N + T131N were responsible for a drastic reduction (>90%) in the quantity of HBsAg. Five out of seven patients whose HBV isolates displayed additional NLG sites remained HBsAg negative despite reactivation. It was concluded that those NLG sites hampered HBsAg recognition and quantification without affecting HBsAg release. Colson et al. [68] found additional NLG sites in two patients experiencing reactivation, T116N and Y134N, with the latter being associated with HBsAg negativity at the time of reactivation. Two more studies reported additional NLG sites in HBsAg-negative patients with reactivation: At position 117 (S117N) [83] and 123 (T123N) [81]. In the light of the given evidence, it was suggested that NLG sites could be crucial in viral evasion from the humoral immune response by masking B-cell epitopes [67,97]. It was shown by a study in vitro [97] that mutants with additional NLG sites had better enveloping and secretion capacity than a wild type virus, thus NLG sites might provide an advantage in replication capacity.

#### 2.1.3. MHR Mutations outside “*a*” Determinant

Although the majority of HBsAg mutations, reported in reactivated HBV, resided within the “*a*” determinant, some were found within the MHR upstream or downstream of the “*a*” determinant sequence. Mutations T118K and P120A are known to alter the antigenicity and recognition by monoclonal anti-HBs antibodies [58,98]. Those mutations are nearby the “*a*” determinant and can disturb the secondary structure of the two loops of amino acids [93]. The mutation at position 122 could be of particular importance because it is responsible for changing the serotype from *adw* to *ayw* or vice versa [99]. Residues 122 and 160 of HBsAg are very important immunologically because they determine the serotype/HBsAg subtype: K122/R122 (*d/y*) and K160/R160 (*r/w*). These serological determinants are mutually exclusive: HBsAg can be *d* or *y* and *r* or *w*. This was discovered very early, shortly after the discovery of HBV serotypes, when it became apparent that *d/y* and *w/r* determinants were involved in cases of anti-HBs and viremia co-existence. It was shown that patients’ sera recognized one of the mutually exclusive determinants while the circulating virus reacted with antibodies against another [100]. It is noticeable that the 122 mutation, providing *d* to *y* or the opposite change, was more than once seen in combination with a mutation at position 144 (D144A/E) [70,72,74,82]. The importance of this pattern was shown in the study by Martel et al. [70], which used site-directed mutagenesis to determine how each of these mutations and their combinations affected recognition by antibodies. They determined that the introduction of the single mutation 122 or 144 could markedly reduce recognition while their combination completely abolished recognition of the modified HBsAg.

#### 2.1.4. Mutations outside MHR

Mutations found outside MHR (Figure 3), in the *N*-terminal and *C*-terminal region of the S domain, mostly affected T-cell epitopes, class I (at positions 41, 44, 48, 93, 96, 97, 175, 176, 178, 185, 190, 207, 213) and II (at position 171) [101]. These changes could also be immune-escape mutations naturally occurring due to the host’s immune surveillance at the T-cell level. Although any severe impact of these mutations has not yet been demonstrated, their importance might exist since appropriate reactivity of T-helper cells is a prerequisite for adequate anti-HBs production and also good cytotoxic T-lymphocyte recognition is needed for effective elimination of infected hepatocytes. The *C*-terminal domain is known to play an important role in HBsAg secretion and viral particle assembly. Although so far only one premature stop codon (W223*) was reported in patients suffering HBV reactivation, they may be very important for intracellular retention of HBV surface proteins, which is the event suggested to have a direct cytopathic effect on infected hepatocytes [102,103]. Stop codons can determine the accumulation of truncated HBsAg in the endoplasmic reticulum, thus inducing oxidative stress and in turn a fulminant course of active hepatitis or hepatocytes’ proliferation and tumor formation [104].

### 2.2. Mutations in the Overlapping Reverse Transcriptase Region

Due to the overlapping of envelope and polymerase genes in the HBV genome, HBsAg mutations correspond to mutations in polymerase/reverse transcriptase (RT) and vice versa. This means that immune-escape mutations in HBsAg can lead to an emergence of mutations associated with resistance to nucleos(t)ide analogues but also that patients undergoing prophylaxis therapy can develop resistance and therefore acquire immune-escape mutations in HBsAg. Antiviral drug-associated mutations can lead to three different changes in the *S* gene [105]: Amino acid substitution mutations, nonsense mutations resulting in truncated surface proteins, and silent mutations. It was shown that an effect comparable to vaccine-escape mutant G145R can be achieved when the patient harbors the combination of E164D + I195M that corresponds to the triple mutant of the RT sequence: rtV173L + rtL180M + rtM204V [106,107]. Following the development of lamivudine resistance with typical mutations in the YMDD motif, the mutated HBV strain can also acquire compensatory mutations rtT128N and rtW153Q, which were found to partially restore viral fitness and correspond to well-known immune-escape HBsAg mutations P120T and G145R [106].

Preventive antiviral treatment is recommended for HBsAg-positive patients before immunosuppressive therapy or chemotherapy and all HIV/HBV co-infected patients regardless of CD4 cell count [16]. Regarding HBsAg-negative patients with the presence of only anti-HBc antibodies who are candidates for immunosuppressive therapy or chemotherapy, antiviral prophylaxis is recommended if they are estimated to be at high risk of HBV reactivation. Recommendations for patients presenting with a serological pattern of resolved HBV infection (HBsAg negative/anti-HBs positive) are still debated. Among studies presenting immune-escape mutations in HBV reactivation, only a few reported mutations that were potentially associated with drug resistance. In many cases, patients were not on antiviral prophylaxis due to an HBsAg-negative/anti-HBs-positive serostatus and most studies found RT mutations were not associated with resistance. Only one study [72] reported pattern rtV173L + rtL180M + rtM204V, which conferred resistance to lamivudine, telbivudine, and partly to entecavir in a patient who achieved HBV DNA suppression only after being treated with tenofovir. The study by Colson et al. [68] reported a higher frequency of RT mutations in patients with reactivation compared to the control group, which is in accordance with more frequent HBsAg mutations in this group but only one of the reported RT mutations is known to confer resistance to antivirals. It was mutation at position rt181, which is known to confer resistance to lamivudine and adefovir. However, this mutation is always associated with a corresponding premature stop codon in HBsAg at position 172, which was not mentioned as an immune-escape HBsAg mutation in this study. This premature stop codon is linked with HBsAg intracellular retention and therefore the risk of hepatocellular carcinoma [108]. One more study [81] reported RT mutations potentially associated with drug resistance: rtL80I in a patient who underwent lamivudine prophylaxis and rtA181S and rtV214A in a patient who did not undergo antiviral prophylaxis but harbored several HBsAg immune-escape mutations. In most cases, it is very difficult to distinguish if HBsAg mutations were selected first and are responsible for RT mutations or the prophylactic therapy influenced resistance mutations, which had a consequential influence on HBsAg. Antiviral prophylaxis has been demonstrated to have a protective role in preventing HBV reactivation, but it would be advisable if nucleos(t)ide analogues with a higher barrier for resistance are used.

### 2.3. Mutations in Basal Core Promoter and Precore Regions

In some patients, the immune-control phase of chronic hepatitis is followed by an HBeAg-negative (sometimes called “immune escape-mutant”) phase, which is characterized by a lack of serum HBeAg with detectable anti-HBe antibodies, moderate to high levels of HBV DNA (>2000 IU/mL), and more rapid progression of liver disease. HBeAg-negative hepatitis became a worldwide-known phenomenon [109], which arises from the selection of specific basal core promoter *(BCP)* and precore *(Pre-C)* mutations in the HBV genome. Typical *Pre-C* mutation involved in HBeAg negativity is G1896A, responsible for introducing a stop codon at position 28 in the *Pre-C* region. It was often found to be accompanied by missense mutation G1899A of codon 29. On the other hand, double mutation within the *BCP* region A1762T/G1764A, commonly occurring in HBeAg-negative patients, was observed to suppress the production of *Pre-C* mRNA and therefore lead to a reduction of HBeAg expression. The double mutation itself leads to a two-fold increase in viral replication, but when it is accompanied by mutations at positions 1753, 1766, and 1768, replication can be increased more than eight-fold [110].

So far, the presence of *BCP* and *Pre-C* mutations has not been often investigated in patients with HBV reactivation. The double *BCP* mutation (A1762T/G1764A), as well as their separate presence and presence of an additional *BCP* mutation C1766G, were reported in some cases of reactivation [68,84,86,90]. *Pre-C* mutations G1896A and A1899G were detected [68,69,90] separately or in combination with mutations in the *BCP* and were correlated with a rapid increase of HBV DNA prior to HBV reactivation [90] and fatal outcomes [68]. The presence of *BCP* and/or *Pre-C* mutations was shown before to result in enhanced viral replication. There is evidence that mutations within the *BCP* are responsible for decreased precore mRNA synthesis, but there have been suggestions that at the same time pregenomic RNA synthesis is unaltered or even enhanced [111]. The HBeAg has been presumed to inhibit the encapsidation of the pre-genome and therefore a lack or decrease of this antigen would enhance the reproduction of HBV [112,113]. The fatal outcome is also associated with mutations in both regions since they were shown to confer more vigorous destruction of infected hepatocytes or even a direct viral cytotoxic effect. HBeAg shares epitopes with HBcAg and has been presumed to serve as a decoy, protecting hepatocytes from immune attack [114,115]. Therefore, the absence of a decoy may be responsible for a more vigorous immune response to core epitopes presented on hepatocytes, even in the setting of immunosuppression or at the time of immune restoration. Also, *BCP* and *Pre-C* mutations can cause a retention of HBcAg in the nucleus of infected hepatocytes, which is speculated to have a direct cytopathic effect [116,117].

## 3. Summary and Conclusions

In the setting of immunosuppression, the impaired balance between viral replication and immune control is responsible for an increase of HBV replication in chronically infected patients or reactivation of inactive HBV in recovered patients. Thus, immunosuppression-related HBV reactivation can occur in a large variety of serological profiles. The highest risk for reactivation was observed in the HBsAg-positive status while the presence of significant levels of anti-HBs antibodies was identified as a protective factor for reactivation development. However, recent studies and case reports emphasized the presence of a high degree of *S* gene variability in HBV isolates from patients who developed reactivation and who in the majority of cases were HBsAg negative with or without anti-HBs antibodies. In reactivated HBV, the most frequently detected mutations belonged to the second loop of the “*a*” determinant in HBsAg. These mutations were previously known to be immune escape and responsible for vaccine- and diagnostic-escape phenomena. Their emergence clearly provides survival in the presence of a developed humoral immune response and is often associated with impaired serological diagnosis of HBV reactivation.

With the current data, it is not possible to determine whether these immune-escape mutations represent part of the viral template in the hepatocytes prior to reactivation and are responsible for assisting in the viral reactivation process or that they emerge during the immunosuppressed state, which allows the development of many quasispecies, including escape mutants. However, the knowledge of their existence strongly supports the use of HBV DNA detection in the monitoring all patients with a history of HBV infection who are undergoing immunosuppression. Since the presence of HBsAg immune-escape mutations can increase the emergence of resistance to antivirals, the use of antiviral drugs with a high genetic barrier to resistance should be recommended for prophylaxis in patients at risk for reactivation.

While the exact role of immune-escape mutations in the pathogenesis of HBV reactivation is unknown at this time, further investigation of their presence and frequency in patients undergoing reactivation will undoubtedly improve the understanding of the process of reactivation and help in the design of protocols for the monitoring and prophylactic therapy of these patients.

## Figures and Tables

**Figure 1 viruses-11-00778-f001:**
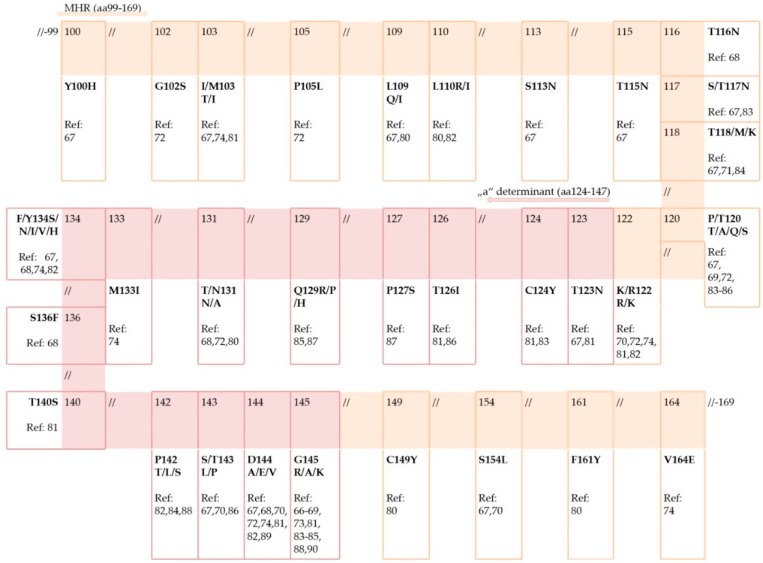
HBsAg mutations within major hydrophilic region (MHR) reported in patients suffering from HBV reactivation; in orange—MHR, aa 99–169; in pink—“*a*” determinant within MHR, aa124–147; Ref—reference [66,67,68,69,70,71,72,73,74,80,81,82,83,84,85,86,87,88,89,90].

**Figure 2 viruses-11-00778-f002:**
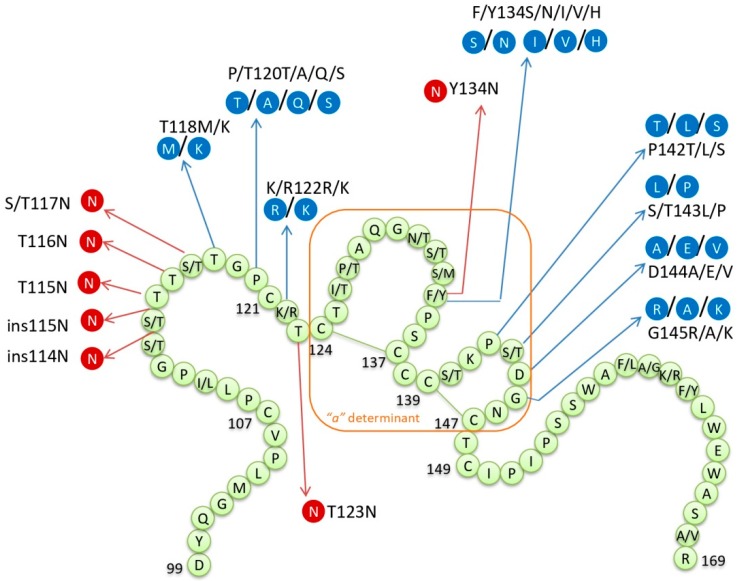
Major hydrophilic region (MHR) of HBsAg with the most frequent mutations and additional *N*-linked glycosylation (NLG) sites found in reactivated HBV; orange rectangle—“*a*” determinant within MHR; in blue—the most frequent mutations; in red—additional NLG sites.

**Figure 3 viruses-11-00778-f003:**
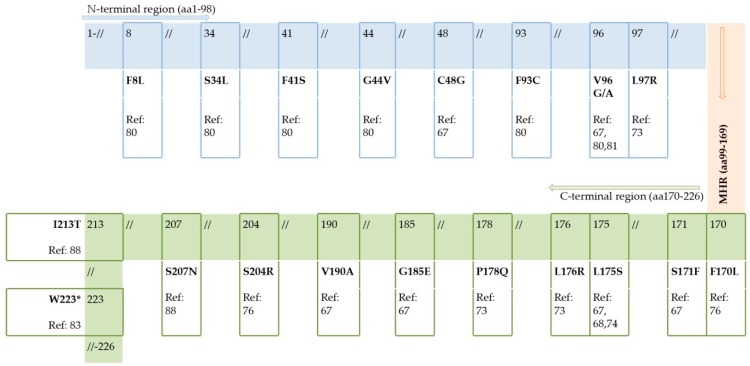
HBsAg mutations outside the major hydrophilic region (MHR) reported in patients suffering from HBV reactivation; in blue—*N*-terminal region of HBsAg, aa1–98; in orange—MHR, aa99–169; in green—C-terminal region of HBsAg, aa170–226; Ref—reference [67,68,73,74,76,80,81,83,88].

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
