# Peer review of "Immune-Escape Hepatitis B Virus Mutations Associated with Viral Reactivation upon Immunosuppression"

_viruses, 2019, doi:10.3390/v11090778_

Round 1
Reviewer 1 Report
In this review, the authors summarized some of the most frequently found hepatitis B virus mutations that are responsible for viral immune-escape and associated with viral reactivation upon immunosuppression. Overall, this review is well-written and has a very clear logic flow in presenting related information. The three illustrations are nicely presented to summarize the mutations found.
There are several minor points that may be addressed by further discussion and clarification
As briefly mentioned by the authors, HBV reactivation may be triggered by direct-acting antiviral therapy for HCV infection in HBV/HCV co-infected patients. This is becoming a very interesting topic in recent years given the highly efficient and specific direct-acting antivirals (DAAs) against HCV. For these cases, it is not very clear whether the HBV reactivation is caused by the HBV mutations discussed in this review. But one theory is that when a patient is infected with HCV, the patient has a very active immune system. When the patient is cured of HCV, his or her immune system is not suppressed any more by HCV and becomes less active, meaning that HBV may then be able to supersede the patient’s immune control. Therefore, it might be better if this review could briefly discuss these co-infected cases. There are some common risk factors for HBV reactivation. For example, if a patient is undergoing chemotherapy or immunosuppression and is HBsAg-positive, especially if the patient is HBV DNA-positive, the patients is at relative high risk of reactivation. In patients who are HBsAg-negative but anti-HBc-positive, the risk of reactivation is much lower. The authors may include a paragraph to discuss these common risk factors for HBV reactivation. Some sentences might be better supported if references are provided. For example, the first sentence talking about the total two billon people infected needs references.
Author Response
Response to Reviewer 1 Comments
Point 1: As briefly mentioned by the authors, HBV reactivation may be triggered by directacting antiviral therapy for HCV infection in HBV/HCV co-infected patients. This is becoming a very interesting topic in recent years given the highly efficient and specific direct-acting antivirals (DAAs) against HCV. For these cases, it is not very clear whether the HBV reactivation is caused by the HBV mutations discussed in this review. But one theory is that
when a patient is infected with HCV, the patient has a very active immune system. When the patient is cured of HCV, his or her immune system is not suppressed any more by HCV and becomes less active, meaning that HBV may then be able to supersede the patient’s immune control. Therefore, it might be better if this review could briefly discuss these co-infected cases.
Response 1: The paragraph referring to the problem of HBV/HCV co-infection and HBV reactivation was added to Introduction (paragraph 1.1).
Point 2: There are some common risk factors for HBV reactivation. For example, if a patient is undergoing chemotherapy or immunosuppression and is HBsAg-positive, especially if the patient is HBV DNA-positive, the patients is at relative high risk of reactivation. In patients who are HBsAg-negative but anti-HBc-positive, the risk of reactivation is much lower. The authors
may include a paragraph to discuss these common risk factors for HBV reactivation.
Response 2: The paragraph about common risk factors for HBV reactivation was added to Introduction (paragraph 1.1)
Point 3: Some sentences might be better supported if references are provided. For example, the first sentence talking about the total two billon people infected needs references.
Response 3: The required references were provided at the end of the sentence.
Reviewer 2 Report
Manuscript Critique
The review entitled “Immune-escape hepatitis B virus mutations associated with viral reactivation upon immunosuppression” focuses on an interesting issue for virologist and hepatologist researchers worldwide, particularly as it relates to monitoring of patients with a history of HBV infection who are undergoing immunosuppression. Although the actual process of immune-escape mutations in pathogenesis is unknown, i.e., pre-existing viral mutations exist prior to reactivation or they emerge as part of the immunosuppression state permitting development of quasi-species, including escape mutants, or a combination of the two, the article adequately addresses the current state of the science and clinical observations. The article highlights the importance of using HBV DNA detection in monitoring of patients with a history of HBV infection and who are undergoing immunosuppression to more accurately screen individuals who otherwise may be misdiagnosed in the absence of detectable HBsAg due to these mutations with or without anti-HBs antibodies.
All of the sections in the review thoroughly the topic areas although several of the sections need editing and/or additional references as follows:
Lines 53 – need reference after “…also been described (ref). Lines 81-83 – rewrite as follows: “It was also reported that any modulation to the immune system that may disrupt the interaction between virus and host could be responsible for HBV reactivation. For instance, it was observed…” Line 92 – rewrite as follows: “The human HBV is a prototypical member of the family…” Line 98 – need reference at end of sentence. Line 113 – it would be helpful to describe what HBeAg is in the context of its pre-core/core transcription. Line 127 – rewrite as follows: “There is evidence an infected hepatocyte will harbour cccDNA and replication intermediates, such as pregenomic RNA (pgRNA), indefinitely until its elimination.” Line 132 – rewrite as follows: “Soon after the discovery of “Australian antigen” or HBsAg, it became apparent that the newly discovered virus was highly variable.” After this sentence, it would be helpful to discuss the extent of the variability and reference. Lines 136-138 – rewrite as follows: “These spontaneous variations are the result of a unique viral life cycle which includes the activity of an error-prone reverse transcriptase.” Line 155 – rewrite as follows: “Since LHB is encoded by all three domains of S-ORF, it has extra amino acids…relative to MHB. LHB is abundant…” Line 183 – rewrite as follows: “When amino acid sequence with “a” determinant is altered, as a result of…” Line 192 – need reference at end of sentence. Line 194 – reference 40 should go at end of sentence. Lines 209-210 – “A significant number of S gene mutations within and outside of “a” determinant have been identified and associated with OBI as follows: …” Figure 1 – needs to have a hatch mark (#) between aa 154 and 161. Line 345 – rewrite as follows: “…to determine how each of these mutations…”
Author Response
Response to Reviewer 2 Comments
Point 1: Lines 81-83 – rewrite as follows: “It was also reported that any modulation to the immune system that may disrupt the interaction between virus and host could be responsible for HBV reactivation. For instance, it was observed…” Line 92 – rewrite as follows: “The human HBV is a prototypical member of the family…”Line 127 – rewrite as follows: “There is evidence
an infected hepatocyte will harbour cccDNA and replication intermediates, such as pregenomic RNA (pgRNA), indefinitely until its elimination.” Line 132 – rewrite as follows: “Soon after the discovery of “Australian antigen” or HBsAg, it became apparent that the newly discovered virus was highly variable.”Lines 136-138 – rewrite as follows: “These spontaneous variations are the result of a unique viral life cycle which includes the activity of an error-prone reverse
transcriptase.” Line 155 – rewrite as follows: “Since LHB is encoded by all three domains of SORF, it has extra amino acids…relative to MHB. LHB is abundant…” Line 183 – rewrite as follows: “When amino acid sequence with “a” determinant is altered, as a result of…”Lines 209-210 – “A significant number of S gene mutations within and outside of “a” determinant have
been identified and associated with OBI as follows: …”Line 345 – rewrite as follows: “…to determine how each of these mutations…”
Response 1: All lines were rewritten as advised.
Point 2: Lines 53 – need reference after “…also been described (ref). Line 98 – need reference at end of sentence. Line 192 – need reference at end of sentence. Line 194 – reference 40 should go at end of sentence.
Response 2: All required references were added or moved to suggested place.
Point 3: Line 113 – it would be helpful to describe what HBeAg is in the context of its precore/core transcription.
Response 3: HBeAg was described as suggested and reference was added.
Point 4: Line 132 – rewrite as follows: “Soon after the discovery of “Australian antigen” or HBsAg, it became apparent that the newly discovered virus was highly variable.” After this sentence, it would be helpful to discuss the extent of the variability and reference.
Response 4: The extent of HBV variability was discussed and references were added.
Point 5: Figure 1 – needs to have a hatch mark (#) between aa 154 and 161.
Response 5: Figure 1 was corrected as advised.